

# The effects of age and central field loss on maintaining balance control when stepping up to a new level under time-pressure

Tjerk Zult[1], Matthew A. Timmis[1,2] and Shahina Pardhan[1]

[1] Vision and Eye Research Institute, School of Medicine, Faculty of Health, Education, Medicine, and Social Care, Anglia Ruskin University, Cambridge, Cambridgeshire, United Kingdom
[2] Cambridge Centre for Sport and Exercise Science, Anglia Ruskin University, Cambridge, Cambridgeshire, United Kingdom

## ABSTRACT

**Objective:** To investigate the effects of age and central field loss on the landing mechanics and balance control when stepping up to a new level under time-pressure.

**Methods:** Eight older individuals with age-related macular degeneration (AMD), eight visually normal older and eight visually normal younger individuals negotiated a floor-based obstacle followed by a 'step-up to a new level' task. The task was performed under (1) no-pressure; (2) time-pressure: an intermittent tone was played that increased in frequency and participants had to complete the task before the tone ceased. Landing mechanics and balance control for the step-up task was assessed with a floor-mounted force plate on the step.

**Results:** Increased ground reaction forces and loading rates were observed under time-pressure for young and older visual normals but not for AMD participants. Across conditions, loading rates and ground reaction forces were higher in young normals compared to older normals and AMD participants. Young visual normals also demonstrated 35–39% shorter double support times prior to and during the step-up compared to older normals and AMD participants. All groups shortened their double support times (31–40%) and single support times (7–9%) in the time-pressure compared to no-pressure condition. Regarding balance control, the centre-of-pressure displacement and velocity in the anterior-poster direction were increased under time-pressure for young and older visual normals but not for AMD participants. The centre-of-pressure displacement and velocity in the medial-lateral direction were decreased for the AMD participants under time-pressure but not for young and older visual normals.

**Conclusions:** Despite walking faster, AMD participants did not adapt their landing mechanics under time-pressure (*i.e.*, they remained more cautious), whilst older and young adults with normal vision demonstrated more forceful landing mechanics with the young being most forceful. A more controlled landing might be a safety strategy to maintain balance control during the step-up, especially in time-pressure conditions when balance control in the anterior-posterior direction is more challenged.

Corresponding author
Tjerk Zult, tjerk.zult@aru.ac.uk

## INTRODUCTION

Falls occur in 30–60% of older adults each year and the majority of the falls are environment-related (*Rubenstein, 2006*). Stepping up or down to a new level is considered the most challenging and hazardous activity of daily living, evidenced by reports that stair falls account for more than 10% of fatal fall accidents (*Startzell et al., 2000*). Although the causes of falls are multifactorial, vision loss has been identified as an important individual risk factor (*Rubenstein, 2006*). Age-related macular degeneration (AMD) is the leading cause of permanent vision loss in the United Kingdom and is characterised by central field loss (*Bunce & Wormald, 2006*). Individuals with AMD report difficulties in performing everyday tasks such as navigating steps and pavement curbs (*Taylor et al., 2016*). They also perceive higher levels of anxiety during scenes that show stair negotiation (*Taylor et al., 2019*) and are twice as likely to experience a fall compared to those with normal vision (*Szabo et al., 2010*; *Soubrane et al., 2007*).

The negotiation of steps and stairs places a high demand on the somatosensory, visual, and vestibular systems (*Startzell et al., 2000*) and the functioning of these systems deteriorates with ageing (*Goble et al., 2009*; *Li et al., 2015*; *Hsieh, Lin & Lee, 2014*; *Congdon et al., 2004*; *Baloh, Jacobson & Socotch, 1993*). The age-related deterioration of these systems is associated with impaired balance control (*McChesney & Woollacott, 2000*; *Aartolahti et al., 2013*; *Lord, 2006*; *Serrador et al., 2009*) and it is widely accepted that poor balance control increases the susceptibility to falling in the elderly (*Muir et al., 2013*; *Mignardot et al., 2014*). Balance control is further deteriorated in older adults with AMD, which resulted in a higher falls risk compared to older adults with normal vision (*Szabo et al., 2008*). The mechanics of stair and step negotiation have been well described for healthy older adults (*Jacobs, 2016*) but studies in older adults with AMD are scant.

To date, only two studies have investigated the movement mechanics in individuals with AMD when stepping up or down to a new level (*Alexander et al., 2014*; *Timmis et al., 2014*). These studies demonstrated that when stepping up to a new level, individuals with AMD adopted a slower (and by implication more cautious) gait speed compared to those with normal vision. Whilst a slower gait speed allows more time to acquire visual information for the control of adaptive locomotion (*Patla & Greig, 2006*), it also results in a longer time spent in a dynamically unstable situation (*i.e.*, single support phase) during step ascent. This longer single support phase might be detrimental for individuals with AMD as their balance control is impaired compared to older adults with normal vision (*Wood et al., 2009*). However, *Wood et al. (2009)* measured balance control during standing conditions instead of dynamic situations such as step ascent.

Two key concepts for balance control are the individual's centre-of-mass (CoM) and centre-of-pressure (CoP) (*Winter, 1995*). The CoM is the point equivalent of the total body mass in the global reference system and is a passive variable controlled by the balance control system (*Winter, 1995*). The CoP is the point location of the vertical ground reaction force

vector and represents the sum of all forces acting between the human body and its supporting surface (*Winter, 1995*). The trajectory of the CoP is commonly evaluated because it reflects the responses of the central nervous system to movements of the CoM.

Research using simulated blur has revealed deficits in balance control when negotiating a step (*Buckley et al., 2005b*). Healthy older adults with simulated blur demonstrated an increased root mean square (change in position of the CoP) in the medial-lateral (ML, sideways) direction when stepping up a curb and the root mean square was further increased when stepping down (*Buckley et al., 2005b*). These findings indicate that ML balance during step ascent was more challenged when vision was blurred. On the other hand, *Heasley et al. (2004)* found that healthy older adults reduced their ML displacement of the CoP (*i.e.*, increased dynamic stability) when stepping up a curb with simulated blur, suggesting that these individuals were able to increase dynamic stability (*i.e.*, improving balance control), likely by adjusting their movement mechanics during step negotiation. They increased their dynamic stability by keeping the horizontal position of the CoM close to the centre of the base of support (*Heasley et al., 2004*). It is possible that adjustments to both postural perturbations and sensory loss are age-dependent, meaning that healthy young adults might react differently to simulated blur compared to healthy older adults when stepping up a curb. This needs to be investigated in future studies. In addition, simulated blur of the whole visual field is different from the loss of the central field and consequent development of an eccentric locus for fixation that often occurs over time in individuals suffering from AMD (*Verghese, Vullings & Shanidze, 2021*).

In all the aforementioned literature on simulated blur (*Buckley et al., 2005b*; *Heasley et al., 2004*) and AMD (*Alexander et al., 2014*; *Timmis et al., 2014*), the negotiation of a step was not time-constrained. However, routine stepping tasks, such as stepping up or down a sidewalk when crossing the road, are often executed under time-pressure (*e.g.*, a traffic signal that restricts the time to cross). To date, it remains unclear how individuals with central field loss negotiate a step in a time-pressure situation.

Step negotiation under time-pressure will limit the time available to view key features in the environment (*e.g.*, curb edge and 3-D dimensions of the curb). Additionally, processing of posture during locomotion becomes more cognitively controlled with ageing and thus requires more of the limited attentional resources (*Ruffieux et al., 2015*), especially when vision is impaired (*Krishnan, Cho & Mohamed, 2017*; *Turano, Geruschat & Stahl, 1998*). Consequently, task-relevant processes other than postural control, such as the awareness of hazards in the environment, become less of a priority (*Young et al., 2016*; *Uiga et al., 2015*; *Ellmers et al., 2016*; *Ellmers & Young, 2019*). It is therefore not surprising that older adults with normal vision and those with AMD adopt a more cautious movement strategy to safely negotiate stairs and steps (*Telonio et al., 2014*; *Novak et al., 2016*; *Christina & Cavanagh, 2002*; *Timmis et al., 2014*; *Alexander et al., 2014*).

Increased walking speed in a time-pressure situation will also affect the mechanics of step and stair negotiation. To illustrate, stair negotiation at an increased walking speed resulted in greater peak joint moments (*i.e.*, turning forces) and powers (*i.e.*, turning forces multiplied by turning velocity) at the ankle, knee, and hip of healthy young adults (*Vallabhajosula, Yentes & Stergiou, 2012*; *Vallabhajosula et al., 2012*). These data suggest

that the lead foot landed on the step with more force and support previous work in healthy young adults who increased the ground reaction forces for the propulsion and stability during locomotion at higher speeds (*Nilsson & Thorstensson, 1989*). Currently, it is unclear how ageing and visual impairment impact step and stair negotiation at increased walking speed.

Therefore, the present study examines the effects of aging and vision loss (*i.e.*, AMD) on balance control and movement kinematics when ascending a step under time-pressure. It is hypothesised that in the no-pressure condition, older adults with AMD will negotiate the step more cautiously (*i.e.*, longer single and double support times, lower ground reaction forces, and slower rates of force development) to maintain balance compared to healthy older adults and young adults. Under time-pressure, a cautious strategy of lengthening the single and double support times can no longer be applied and increased ground reaction forces and rates of force development are necessary to maintain balance. However, older adults will show less of an increase (*i.e.*, remain more cautious) to an extent that balance is still well controlled, and this effect will be exaggerated when central vision is impaired due to AMD.

## MATERIALS AND METHODS

### Participants

A total of 24 participants were recruited to take part in this study. Eight were diagnosed with AMD by an ophthalmologist, eight were healthy older adults with normal or corrected-to-normal vision, and eight were healthy young adults with normal or corrected-to-normal vision. Inclusion criteria were: ability to walk without a walking aid, aged ≥60 years for AMD participants and healthy older adults, aged 18–35 years for healthy young adults, visual impairment (AMD participants only) as defined by the World Health Organization International Classification of Diseases (*i.e.*, visual acuity >0.301 logMAR) (*World Health Organization, 2018*). Exclusion criteria were: any neuromuscular and skeletal problems that could interfere with balance or gait, usage of medication that causes dizziness, and worse than normal vision for healthy older and young adults (*i.e.*, visual acuity >0.301 logMAR) (*World Health Organization, 2018*). The health of the participants was assessed through a self-report questionnaire and subjective physical activity levels were assessed in a face-to-face interview using the Global Physical Activity Questionnaire (*World Health Organization, 2012*). The experimental procedures were in accordance with the Declaration of Helsinki and written informed consent was obtained from all participants prior to the start of the study. Study approval was provided by the Research Ethics Committee of the Anglia Ruskin University (approval number: FMSFREP 16/17 008).

### Visual examination

The visual examination included three tests that were executed in the same order for all participants while wearing best-corrected spectacles. First, binocular visual acuity was measured using the Bailey-Lovie logMAR chart at a working distance of 4 m using a letter-by-letter scoring system (0.02 logMAR) (*Bailey & Lovie, 1976*). For AMD participants, this

was their measured visual acuity whether they used eccentric fixation or not. Shorter distances were used when a subject was not able to read the largest size letters at 4 m distance and scores were adjusted accordingly. Second, binocular contrast sensitivity was assessed using the Pelli-Robson chart at 1 m distance, and scored per group of three letters (0.15 log units) of which two had to be correct (*Pelli & Robson, 1988*). Third, the monocular visual field was examined with a Humphrey Field Analyzer SITA-Standard 30-2 threshold test (Carl Zeiss Meditec Inc., Dublin, CA, USA). The "best location" model was used to calculate the binocular visual field from the monocular visual field scores (*Nelson-Quigg, Cello & Johnson, 2000*).

## Experimental setup

The walkway was 7 m long, 1.2 m wide, and positioned in the middle of a research laboratory. The view of the travel path was blocked at the start of each trial by a cardboard wall. Participants were instructed to place their right hand on a pressure pad before the start of each trial. Each trial commenced after the sound of a single 'beep'. Then, participants released their hand from the pressure pad, turned round the corner of the cardboard wall, and started walking along the travel path. The travel path consisted of a 10.0 cm high, 62.0 cm wide, and 1.8 cm deep obstacle followed by a floor-mounted force plate (4.5 cm high, 50.2 cm wide, and 50.2 cm deep) reflecting a pavement curb to step-up onto at a pedestrian crossing point. A step-up height of 4.5 cm was chosen because curb heights lower than 6.0 cm are particularly difficult for blind and partially sighted to detect the edge of the curb (*Thomas, 2011*). Participants had to step over the obstacle followed by the step-up onto the floor-mounted force plate. The obstacle was included to ensure that participants did not simply target the single step at the end of the walkway, rather, had to initially attend to the obstacle. The trial was completed after the right hand was placed on the pressure pad at the end of the walkway after stepping up onto the step. Another cardboard wall at the end of the walkway prevented the participants from seeing the data collectors. The pressure pads were used to record the duration of each trial. The obstacle was randomly positioned between 4.05 and 5.30 m from the start position and the force plate was always positioned at 7.00 m from the start position. Therefore, participants had to adjust their gait between trials which prevented the learning of a repeated motor pattern. The height of the obstacle and force plate reflected typical heights encountered in everyday life (*Timmis & Pardhan, 2012*). The colour of the obstacle and force plate contrasted with the black background of the laboratory carpet.

All participants performed the task with their best-corrected vision. Participants started the experiment with nine baseline walking trials in which no obstacle was present (*i.e.*, to get familiar with the task and walking surface). They had to walk up to the force plate at their comfortable walking speed and step onto the force plate before placing the hand on the pressure pad at the end of the walkway (*i.e.*, to prevent the participants from having extra support during the step-up phase). These trials were followed by trials where they had to negotiate a floor-based obstacle before stepping onto the force plate under a time-pressure and no-pressure condition. Time-pressure was induced by a custom-made timing device that produced an intermittent tone that increased in frequency as a function

of the total trial duration. The intermittent tone started when the hand was released from the pressure pad at the start and stopped when the hand was placed on the pressure pad at the end of the walkway. Participants were instructed to finish the task before the tone extinguished which meant that they had to walk 20% faster than their comfortable walking speed throughout the entire trial. A 20% faster walking speed was deemed feasible for both young and older participants (*Sheik-Nainar & Kaber, 2007*). The participants heard the intermittent tone once before starting the time-pressure trials so they were familiar with the duration of each trial. Participants' comfortable walking speed was calculated as the average walking speed over the last three baseline walking trials. In the no-pressure condition, the tone was absent and participants were told to perform the task at their comfortable walking speed. Four trials were performed per condition of which three trials included the obstacle and one trial was without the obstacle. Before the start of each trial, participants did not know whether the obstacle would be present or not. The condition was known to the participants and practice trials were not permitted. Participants randomly started with the no-pressure or time-pressure condition.

## Data collection

Prior to data collection, 14 reflective markers were attached to the following body locations either to the shoes or directly to the skin: hallux, fifth metatarsal head, medial and lateral side of the posterior part of the calcaneus, nail of the thumb, nail of the index finger, and the ulnar styloid process. In addition, four markers were placed on the edges of the pressure pad at the end of the walkway to determine when the hand touched the pad (denoting the end of a trial), and two markers each were placed on the upper front edge of the obstacle and force plate to determine the height and location of both hazards within the laboratory coordinate system.

An eight-camera 3-D motion capture system (Vicon, Hauppauge, NY, USA; Oxford Metrics Ltd, Oxford, UK) was used to record the marker trajectories at 100 Hz during the performance of the adaptive gait task. The analysis of the marker data was performed in Visual 3D (C-Motion Inc., Rockville, MD, USA). The data were filtered with a fourth-order low-pass Butterworth filter at 7 Hz before timestamps were calculated of adaptive gait events.

A single force plate (AccuGait; Advanced Mechanical Technology, Inc., Watertown, MA, USA) was used to collect the ground reaction forces and CoP data at 100 Hz. Data were recorded with the NetForce Acquisition Software (version 2.4.0; Advanced Mechanical Technology, Inc., Watertown, MA, USA). Further analysis of the data was performed with a custom Matlab script (version 2016a; The Mathworks, Natick, MA, USA) in which the data were filtered with a fourth-order low-pass Butterworth filter at 8 Hz before the ground reaction forces and CoP data were calculated.

## Data analysis

The total performance time was measured from the instant that the right hand was released from the pressure pad at the start of the trial until the right hand was placed on the pressure pad at the end of the walkway, denoting the end of the trial.
Additional temporal variables were obtained from analysis of the 3-D marker data in Visual3D (C-motion Inc., Rockville, MD, USA). The following temporal variables were calculated, which have previously been determined as important for stepping up to a new level (*Timmis et al., 2014*):

1. Double support phase before step-up in seconds: time from final foot placement (*i.e.*, instant of heel strike) to lead foot toe-off before stepping onto the force plate.
2. Single support phase during step-up lead foot in seconds: swing time of the lead limb during the step-up onto the force plate whereby only the trail foot is in contact with the ground.
3. Double support phase during step-up in seconds: time from lead foot contact on the force plate to trail foot toe-off before stepping onto the force plate.
4. Single support phase during step-up trail foot in seconds: swing time of the trail limb during the step-up onto the force plate whereby only the lead foot is in contact with the force plate.

Toe-off was defined as the instant where the resultant velocity of the foot's toe marker first increased more than 0.9 m/s for ten consecutive frames (*Zult et al., 2019*). Participants' initial contact with the force plate was either with the heel or forefoot. Heel strike was defined as the instant when the resultant velocity of the foot's medial heel marker first decreased less than 0.6 m/s for ten consecutive frames (*Zult et al., 2019*). Toe strike was defined as the instant when the resultant velocity of the foot's toe marker first decreased less than 0.2 m/s for ten consecutive frames.

Note that Data Analysis S1 includes the analysis of kinematic variables related to obstacle negotiation. The results of this analysis will be published alongside this article because the current article focuses on the 'step-up to a new level' task. The included kinematic variables for obstacle negotiation are similar to a previous article that investigated the effects of time-pressure in participants with (simulated) vision loss (*Zult et al., 2019*).

Figure 1 depicts a schematic representation of the step-up action and the measured ground reaction forces. Peak ground reaction forces were calculated in the vertical ('z'), anterior-posterior ('y'), and medial-lateral ('x') direction during the landing of the lead foot on the force plate and expressed as a percentage of body mass. The loading rate (*i.e.*, rate of force development) was also calculated in each direction from the instant that the lead foot contacted the force plate (*i.e.*, a vertical ground reaction force above 20 N) (*Buckley et al., 2005b*) to the time-point that the force reached its peak. The loading rate is expressed in N/s.

The X and Y coordinates of the CoP were calculated using the formulas provided by the manufacturer of the force plate (Advanced Mechanical Technology, Inc.):

$$x = \frac{-\left(M_y + F_x \times d_z\right)}{F_z} \qquad\qquad y = \frac{-\left(M_z + F_y \times d_z\right)}{F_z}$$

$M_y$ and $M_z$ are the moments in the anterior-posterior and vertical direction respectively. $F_x$, $F_y$, and $F_z$ are the forces in the medial-lateral, anterior-posterior, and vertical direction respectively. $d_z$ is the thickness of the force plate which was 41.3 mm.

Peerj

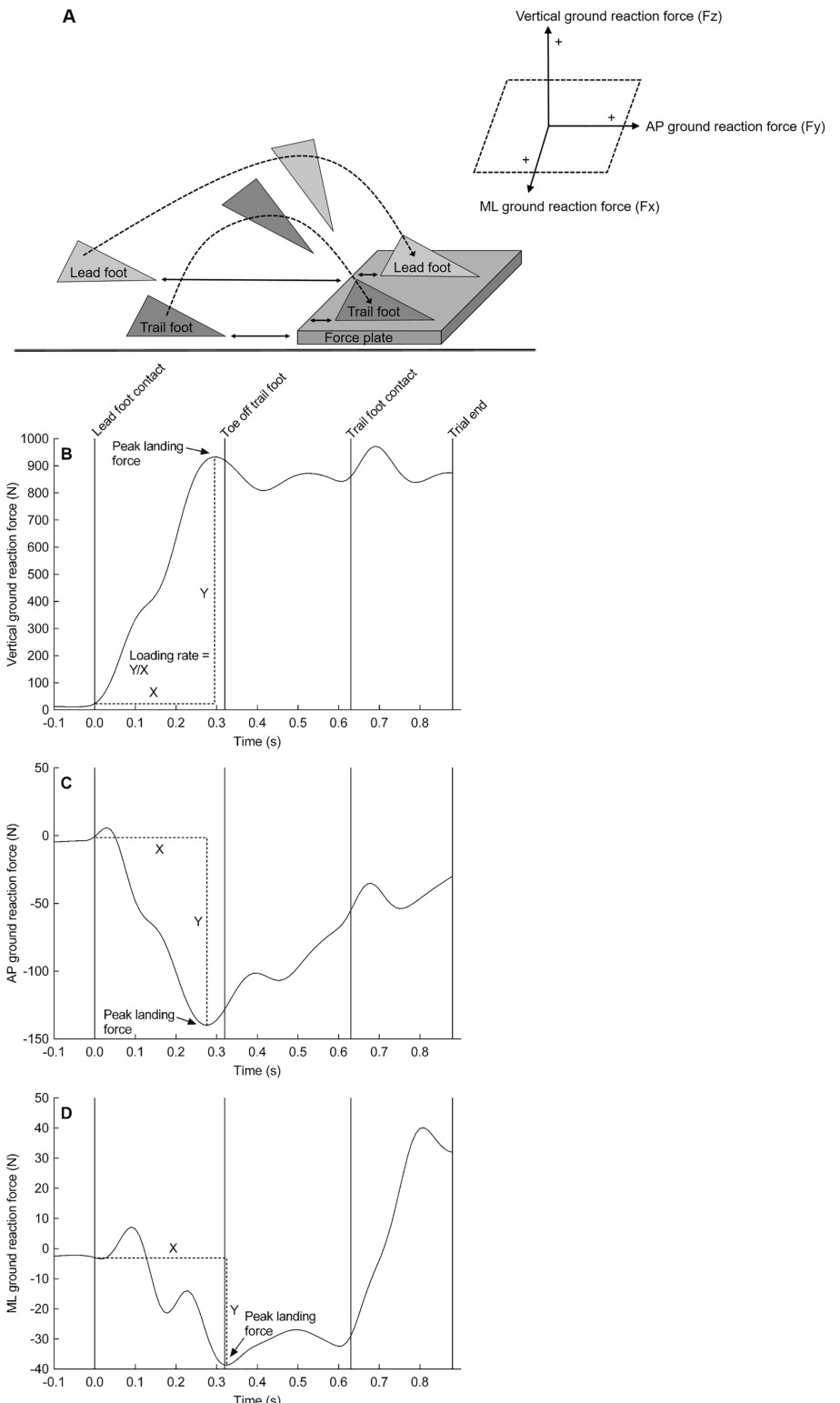

**Figure 1 Experimantal setup and force plate data.** Schematic representation of the step-up action (A) and a representative trace of the ground reaction force from a single participant in the vertical direction (B), anterior-posterior direction (C), and medial-lateral direction (D) when lading onto the floor-mounted force plate. Of note, the left foot was the lead foot in this example. AP, anterior-posterior; ML, medial-lateral.
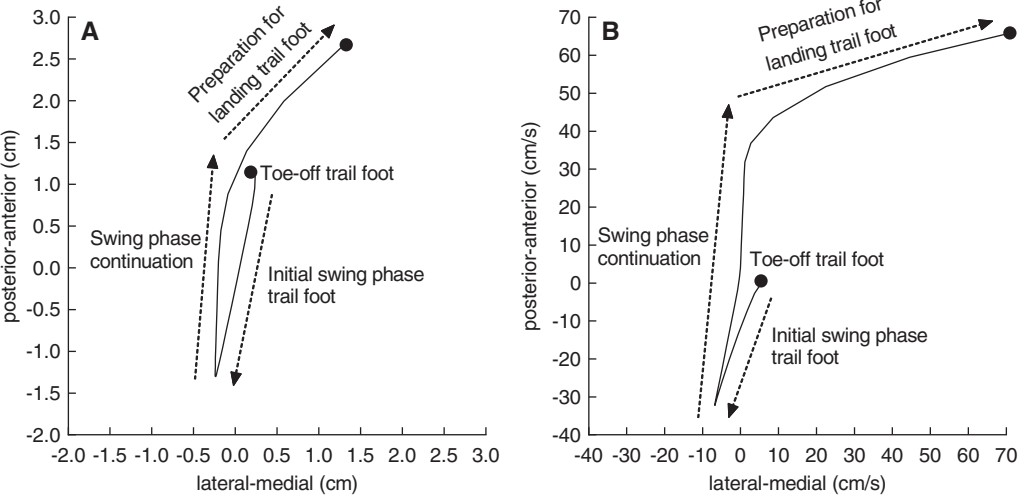

**Figure 2 A representative trace of the centre-of-pressure displacement and velocity.** A representative trace of the centre-of-pressure displacement (A) and velocity (B) from a single participant when stepping onto the floor-mounted force plate. Of note, the left foot was the lead foot in this example. Postural control parameters were calculated from the centre-of-pressure data (see text for details).

Figure 2 shows the CoP data that were used to analyse the main postural control parameters. The formulas for calculating those parameters were adopted from previous literature (*Khanmohammadi et al., 2017*). The following postural control parameters were calculated for the single support phase during the step-up action of the trail foot (kinematic variable 4):

1. Total displacement of the CoP in the anterior-posterior and medial-lateral direction expressed in cm.

2. Average velocity of the CoP displacements in the anterior-posterior and medial-lateral direction expressed in cm/s.

Even though the instructions were to complete the step-up action before placing the hand on the pressure pad (*i.e.*, denoting the end of the trial), some participants placed the hand on the pressure pad before the step-up action was completed in 21% of the trials (29 out of 141 trials). To ensure that placement of the hand on the pressure pad did not bias force plate data (*i.e.*, improving balance control), these trials were removed and instead, we only analysed the first correctly performed trial in each condition.

In the time-pressure condition, seven participants ran out of time, meaning that they performed the trial slower than the pre-defined target time. These included three AMD participants (six trials), two older visual normals (three trials), and two young visual normals (three trials). These 12 trials were not repeated because each participant had at least one successful time-pressure trial. The 12 non-successful time-pressure trials were not included in the final analysis.

**Table 1 Baseline characteristics of the participants.** Baseline characteristics of the participants (mean ± SD).

| | AMD participants ($n = 8$) | Older visual normals ($n = 8$) | Young visual normals ($n = 8$) |
|---|---|---|---|
| Age (years)[‡] | 76 (9) | 70 (4) | 27 (5) |
| Sex (male/female) | 3/5 | 4/4 | 2/6 |
| Mass (kg) | 71 (12) | 68 (12) | 68 (13) |
| Height (cm) | 164 (9) | 169 (10) | 170 (13) |
| Visual acuity (logMAR)[*] | 0.79 (0.15) | 0.02 (0.12) | −0.18 (0.10) |
| Contrast sensitivity (logCS)[*] | 0.62 (0.27) | 1.69 (0.07) | 1.95 (0.00) |
| Visual fields (dB)[†] | 22 (5) | 31 (0) | 32 (1) |
| Sedentary behaviour (min/day)[†] | 2,048 (1,284) | 1,916 (789) | 4,305 (1,706) |
| Moderate-to-vigorous physical activity (min/day) | 826 (867) | 834 (505) | 512 (362) |

Notes:
[*] Significant group difference between each group ($p < 0.05$).
[†] Significant group difference between AMD participants and the other two groups ($p < 0.05$).
[‡] Significant group differences between young visual normals and the other two groups ($p < 0.05$).

## Statistical analysis

Data in text and figures are expressed as mean ± SD. The statistical analysis was performed using SPSS version 24. The Kolmogorov-Smirnov test was used to check normality and each variable was normally distributed. Between group differences in group characteristics were determined with a one-way analysis of variance (ANOVA) for the continuous variables and a Chi-square test for the dichotomous variable sex. A group (AMD, older normals, young normals) by condition (no-pressure, time-pressure) mixed ANOVA was performed to test for differences in total performance time, peak ground reaction forces, loading rates, temporal kinematic variables, and postural control parameters. Significant $F$ values from the ANOVAs were subjected to an LSD *post hoc* pairwise comparison to determine the means that were different. The level of significance ($\alpha$) was set at $p < 0.05$. Effect sizes were calculated using Cohen's $d$.

A priory power analysis with G*Power 3.1 was performed to calculate the sample size that was required to obtain a significant group by condition effect on the ANOVA for each variable of interest. The effects of ageing and visual impairment on step negotiation are evident when there are no time constrains but these effects are unknown for time-constrained situations. Therefore, a medium effect size of 0.50 was used for the power analysis to prevent underestimation of the sample size. The calculated sample size was 21 (*i.e.*, seven subjects per group) based on an effect size of 0.50 with a power of 95% at the $p < 0.05$ significance level.

## RESULTS

### Group characteristics

Table 1 shows the group characteristics. Expected significant differences were found for age ($F_{2, 21} = 147.9$, $p < 0.001$). *Post hoc* testing revealed that young visual normals were significantly younger than older visual normals and AMD participants (both $p < 0.001$, $d \geq 6.73$). There was no significant difference in age between older visual normals and AMD participants ($p = 0.059$).

A significant main effect of group was also observed for visual acuity ($F_{2, 21} = 133.9$, $p < 0.001$), contrast sensitivity ($F_{2, 21} = 152.3$, $p < 0.001$), and visual fields ($F_{2, 21} = 23.7$, $p < 0.001$). *Post hoc* testing showed that visual acuity and contrast sensitivity were better in young visual normals than older visual normals and AMD participants (all $p \leq 0.004$, d $\geq$ 1.81), and were better in older visual normals than AMD participants (both $p < 0.001$, d $\geq$ 5.43). Visual fields were significantly better in young and older visual normals compared to AMD participants (both $p < 0.001$, d $\geq$ 2.55). Visual fields were not significantly different between young and older visual normals ($p = 0.513$). Of note, none of the participants contacted the obstacle or curb edge during the performance of the mobility course.

Sedentary behaviour and moderate-to-vigorous physical activity were assessed with the Global Physical Activity Questionnaire. A main effect of group was observed for sedentary behaviour ($F_{2, 21} = 8.4$, $p < 0.002$) but not for moderate-to-vigorous physical activity ($F_{2, 21} = 0.7$, $p < 0.503$). *Post hoc* analysis showed that young visual normals spent more time per week being sedentary than older visual normals and AMD participants (both $p = 0.002$, d $\geq$ 1.41).

## Total performance time

The total performance time showed a significant main effect of group ($F_{2, 21} = 7.4$, $p = 0.004$) and condition ($F_{1, 21} = 91.1$, $p < 0.001$) (Table 2). *Post hoc* testing for the group effect showed that young visual normals had a 32% shorter total performance time than the AMD participants ($p = 0.001$, d = 1.32). No other significant differences were observed between groups ($p \geq 0.054$). The condition effect revealed that the total performance times was 24% shorter in the time-pressure than no-pressure condition ($p < 0.001$, d = 1.07). No significant group by condition interaction was observed ($F_{2, 21} = 0.5$, $p = 0.616$).

## Normalised peak ground reaction forces

A main effect of group was found for the vertical component of the peak ground reaction force ($F_{2, 21} = 7.6$, $p = 0.003$, Table 2). *Post hoc* testing revealed that the peak vertical ground reaction force in young visual normals was 7–10% higher than older visual normals and AMD participants (both $p \leq 0.013$, d $\geq$ 0.79). No significant condition effect ($F_{1, 21} = 3.3$, $p = 0.083$) or group by condition interaction ($F_{2, 21} = 1.8$, $p = 0.184$) was observed.

The anterior-posterior component of the peak ground reaction force showed a group ($F_{2, 21} = 11.4$, $p < 0.001$), condition ($F_{1, 21} = 12.1$, $p = 0.002$), and interaction effect ($F_{2, 21} = 5.2$, $p = 0.015$) (Table 2). *Post hoc* testing for the interaction effect showed that in the time-pressure condition only, young visual normals had a 35–96% higher peak force than older visual normals and AMD participants (both $p \leq 0.003$, d $\geq$ 1.41) and older visual normals demonstrated a 46% higher peak force than AMD participants ($p = 0.006$, d = 1.50). On comparing between conditions for each group, young and older visual normals showed a 31–39% higher force in the time-pressure *vs.* no-pressure condition (both $p \leq 0.020$, d $\geq$ 0.85), while AMD participants did not significantly change their peak ground reaction force ($p = 0.637$).

**Table 2 Task performance variables of AMD participants, older visual normals and young visual normals in the two different conditions (mean ± SD).**

| Variables | AMD participants ($n$ = 8) | | Older visual normals ($n$ = 8) | | Young visual normals ($n$ = 8) | | ANOVA |
|---|---|---|---|---|---|---|---|
| | No-pressure | Time-pressure | No-pressure | Time-pressure | No-pressure | Time-pressure | |
| Total performance time (s) | 10.0 (2.0) | 7.8 (1.8) | 8.7 (1.9) | 6.5 (1.7) | 7.0 (0.5) | 5.2 (0.7) | G, C |
| Normalised peak ground reaction forces | | | | | | | |
| Peak vertical ground reaction force (% body mass) | 101 (3) | 102 (3) | 104 (8) | 104 (10) | 106 (4) | 116 (13) | G |
| Peak AP ground reaction force (% body mass) | −16 (6) | −15 (3) | −16 (6) | −21 (5) | −21 (4) | −29 (5) | G, C, GxC |
| Peak ML ground reaction force (% body mass) | 6 (1) | 7 (1) | 6 (2) | 6 (1) | 6 (0) | 8 (2) | ns |
| Loading rates | | | | | | | |
| Loading rate in vertical direction (N/s) | 2,128 (508) | 2,224 (554) | 2,107 (764) | 3,642 (1,401) | 3,011 (510) | 4,515 (1,560) | G, C, GxC |
| Loading rate in AP direction (N/s) | −560 (365) | −461 (284) | −431 (347) | −878 (423) | −665 (195) | −1,334 (719) | G, C, GxC |
| Loading rate in ML direction (N/s) | 151 (76) | 134 (49) | 146 (100) | 197 (65) | 177 (49) | 308 (150) | G, C, GxC |
| Temporal variables | | | | | | | |
| Double support phase prior to step-up (s) | 0.19 (0.04) | 0.13 (0.03) | 0.19 (0.06) | 0.12 (0.05) | 0.14 (0.03) | 0.06 (0.03) | G, C |
| Single support phase during step-up lead foot (s) | 0.42 (0.07) | 0.38 (0.05) | 0.45 (0.03) | 0.39 (0.03) | 0.41 (0.03) | 0.41 (0.12) | C |
| Double support phase during step-up (s) | 0.32 (0.09) | 0.26 (0.09) | 0.33 (0.13) | 0.22 (0.09) | 0.23 (0.04) | 0.12 (0.03) | G, C |
| Single support phase during step-up trail foot (s) | 0.31 (0.04) | 0.29 (0.03) | 0.34 (0.03) | 0.31 (0.05) | 0.32 (0.03) | 0.30 (0.05) | C |
| Balance control single support phase | | | | | | | |
| CoP displacement in AP direction (cm) | 2.7 (2.2) | 2.3 (1.0) | 1.9 (1.0) | 3.6 (2.1) | 2.2 (1.0) | 5.0 (1.7) | C, GxC |
| CoP displacement in ML direction (cm) | 3.2 (1.6) | 1.8 (1.4) | 1.0 (0.6) | 1.6 (1.0) | 1.4 (1.1) | 1.5 (0.7) | G, GxC |
| Centre-of-pressure velocity in AP direction (cm/s) | 8.6 (5.8) | 8.6 (4.1) | 5.8 (2.7) | 12.0 (6.5) | 7.1 (3.3) | 17.6 (5.2) | C, GxC |
| Centre-of-pressure velocity in ML direction (cm/s) | 11.5 (6.5) | 6.4 (5.4) | 3.3 (2.4) | 5.4 (3.6) | 4.6 (4.1) | 5.4 (2.1) | G, GxC |

**Note:**
AP, anterior-posterior; CoP, centre-of-pressure; ML, medial-lateral; G, significant group effect ($p < 0.05$); C, significant condition effect ($p < 0.05$); GxC, significant group by condition interaction ($p < 0.05$); ns, no significant effect.

No significant group ($F_{2, 21} = 1.2$, $p = 0.336$), condition ($F_{1, 21} = 2.1$, $p = 0.161$), or interaction effect ($F_{2, 21} = 1.2$, $p = 0.327$) was found for the medial-lateral component of the peak ground reaction force (Table 2).

**Rate of force development**

The rate of force development in the vertical direction showed a group ($F_{2, 21} = 8.2$, $p = 0.002$), condition ($F_{1, 21} = 18.6$, $p < 0.001$), and interaction effect ($F_{2, 21} = 3.8$, $p = 0.038$) (Fig. 3A, Table 2). *Post hoc* testing for the interaction effect showed that young visual normals had a 41–43% higher rate of force development than older visual normals and AMD participants in the no-pressure condition (both $p \leq 0.007$, d ≥ 1.32), and young and older visual normals had a 64–103% higher rate of force development compared to AMD participants in the time-pressure condition (both $p \leq 0.034$, d ≥ 1.26). On comparing between conditions for each group, young and older visual normals showed a 50–73% higher rate of force development in the time-pressure *vs.* no-pressure condition (both $p \leq$

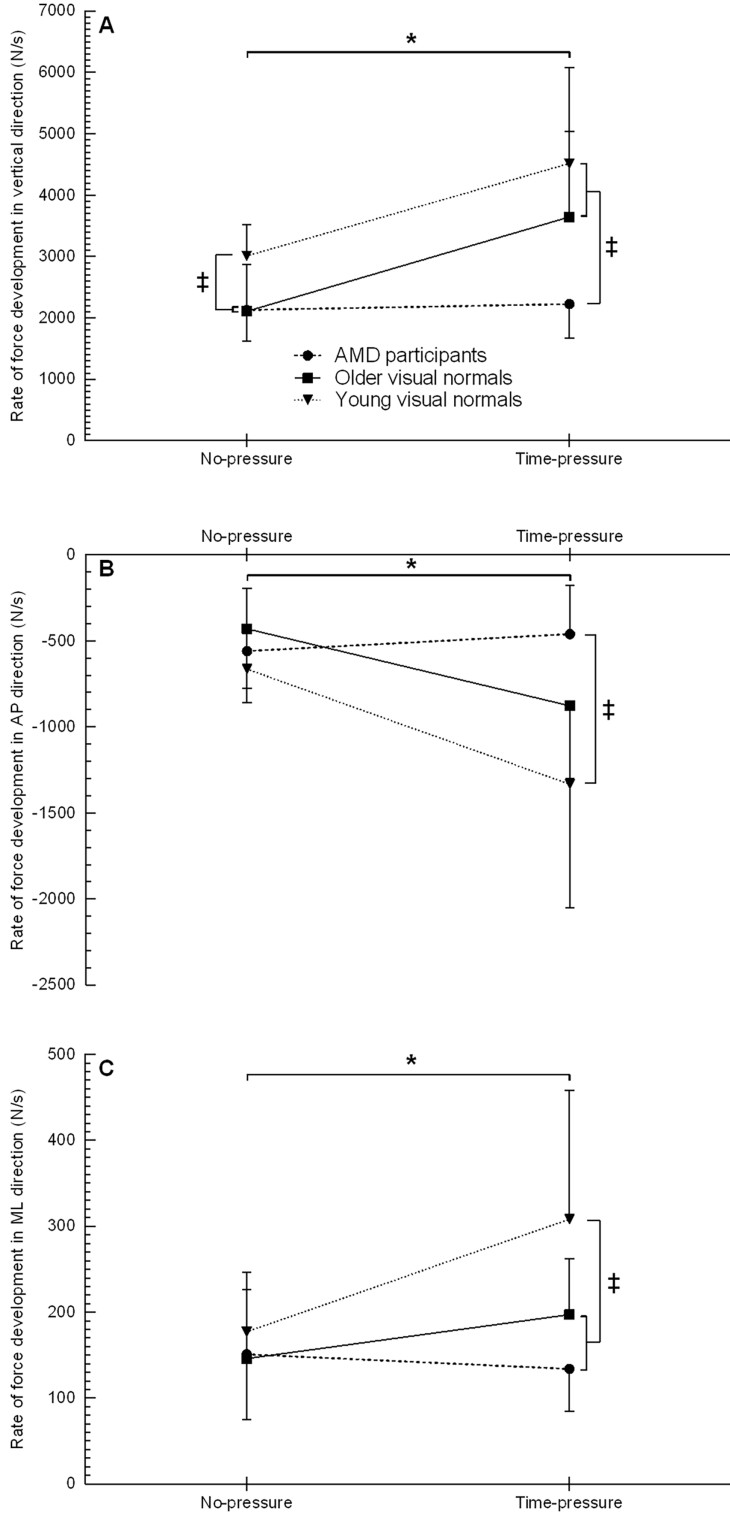

**Figure 3 Rate of force development when stepping up to the force platform with the lead foot in the time-pressure and no-pressure condition.** Rate of force development in the vertical direction (A), anterior-posterior direction (B), and medial-lateral direction (C) in the two different conditions when stepping up to the force platform with the lead foot. ‡, between group difference ($p < 0.05$); *, group by condition interaction between the time-pressure and no-pressure condition ($p < 0.05$).

0.002, d ≥ 1.23), while AMD participants did not significantly change their rate of force development ($p = 0.822$).

The rate of force development in the anterior-posterior direction showed a group ($F_{2, 21} = 4.4$, $p = 0.025$), condition ($F_{1, 21} = 10.9$, $p = 0.003$), and interaction effect ($F_{2, 21} = 4.9$, $p = 0.017$) (Fig. 3B, Table 2). *Post hoc* testing for the interaction effect revealed that in the time-pressure condition only, young visual normals had a 189% higher rate of force development than AMD participants ($p = 0.003$, d = 1.51), while no between group differences were found for older visual normals (both $p ≥ 0.087$). On comparing between conditions for each group, young and older visual normals demonstrated a 101–104% higher rate of force development in the time-pressure *vs.* no-pressure condition (both $p ≤ 0.020$, d ≥ 1.09), while AMD participants did not significantly change their rate of force development ($p = 0.585$).

The rate of force development in the medial-lateral direction showed a group ($F_{2, 21} = 3.8$, $p = 0.040$), condition ($F_{1, 21} = 7.9$, $p = 0.011$), and interaction effect ($F_{2, 21} = 4.7$, $p = 0.020$) (Fig. 3C, Table 2). *Post hoc* testing for the interaction effect revealed that in the time-pressure condition only, young visual normals had a 56–129% higher rate of force development than older visual normals and AMD participants (both $p ≤ 0.036$, d ≥ 0.90), while the rate of force development was not significantly different between older visual normals and AMD participants ($p = 0.215$). On comparing between conditions for each group, young visual normals demonstrated a 74% higher rate of force development in the time-pressure *vs.* no-pressure condition ($p = 0.001$, d = 1.11), while older visual normals and AMD participants did not significantly change their rate of force development (both $p ≥ 0.147$).

### Temporal variables
#### *Duration of the double support phase*
The double support phase prior to the step-up showed a significant main effect of group ($F_{2, 21} = 6.5$, $p = 0.006$) and condition ($F_{1, 21} = 111.9$, $p < 0.001$) (Table 2). *Post hoc* testing for the group effect showed that young visual normals had a 37–38% shorter double support phase than the older visual normals and AMD participants (both $p ≤ 0.007$, d ≥ 1.26). The double support phase was not significantly different between older visual normals and AMD participants ($p = 0.818$). The condition effect revealed that the double support phase was 40% shorter in the time-pressure than no-pressure condition ($p < 0.001$, d = 1.40). No significant group by condition interaction was observed ($F_{2, 21} = 1.0$, $p = 0.394$).

The double support phase during the step-up showed a significant main effect of group ($F_{2, 21} = 4.7$, $p = 0.021$) and condition ($F_{1, 21} = 86.8$, $p < 0.001$) (Table 2). *Post hoc* testing for the group effect showed that young visual normals had a 35–39% shorter double support phase than the older visual normals and AMD participants (both $p ≤ 0.025$, d ≥ 1.12). The double support phase was not significantly different between older visual normals and

AMD participants ($p$ = 0.666). The condition effect revealed that the double support phase was 31% shorter in the time-pressure than no-pressure condition ($p$ < 0.001, d = 0.94). No significant group by condition interaction was observed ($F_{2, 21}$ = 2.8, $p$ = 0.084).

*Duration of the single support phase*
The single support phase, during the step-up of the lead foot, showed a significant main effect of condition ($F_{1, 21}$ = 6.0, $p$ = 0.024, Table 2). *Post hoc* testing for the condition effect revealed that the single support phase was 9% shorter in the time-pressure than no-pressure condition ($p$ = 0.024, d = 0.57). No significant group ($F_{2, 21}$ = 0.4, $p$ = 0.707) or interaction effect ($F_{2, 21}$ = 1.5, $p$ = 0.255) was observed.

The single support phase, during the step-up of the trail foot, showed a significant main effect of condition ($F_{1, 21}$ = 5.4, $p$ = 0.031, Table 2). *Post hoc* testing for the condition effect showed that the single support phase was 7% shorter in the time-pressure than no-pressure condition ($p$ = 0.031, d = 0.57). No significant group ($F_{2, 21}$ = 1.5, $p$ = 0.256) or interaction effect ($F_{2, 21}$ = 0.2, $p$ = 0.826) was observed.

## Balance control single support phase
### Centre-of-pressure displacement
The CoP displacement in the anterior-poster direction showed a condition ($F_{1, 21}$ = 8.9, $p$ = 0.007) and interaction effect ($F_{2, 21}$ = 4.1, $p$ = 0.032) (Fig. 4A, Table 2). *Post hoc* testing for the interaction effect showed that in the time-pressure condition only, young visual normals had 115% more CoP displacement compared to AMD participants ($p$ = 0.04, d = 1.81). No other between group differences were observed (all $p \geq$ 0.111). On comparing between conditions for each group, young and older visual normals showed 89–128% more CoP displacement in the time-pressure *vs.* no-pressure condition (both $p \leq$ 0.045, d $\geq$ 0.98), while AMD participants did not significantly change their CoP displacement ($p$ = 0.647). No significant main effect of group was observed ($F_{2, 21}$ = 2.0, $p$ = 0.158).

The CoP displacement in the medial-lateral showed a group ($F_{2, 21}$ = 4.6, $p$ = 0.023) and interaction effect ($F_{2, 21}$ = 4.7, $p$ = 0.025) (Fig. 4B, Table 2). *Post hoc* testing for the interaction effect showed that in the no-pressure condition only, young and older visual normals had 56–68% less CoP displacement than AMD participants (both $p \leq$ 0.005, d $\geq$ 1.27). No other between group differences were observed (all $p \geq$ 0.511). On comparing between conditions for each group, AMD participants decreased their CoP displacement 46% in the time-pressure *vs.* no-pressure condition ($p$ = 0.009, d = 0.95), while young and older visual normals did not significantly change their CoP displacement (both $p \geq$ 0.276). No significant main effect of condition was observed ($F_{1, 21}$ = 0.9, $p$ = 0.367).

### Centre-of-pressure velocity
The average CoP velocity in the anterior-posterior direction showed a condition ($F_{1, 21}$ = 17.5, $p$ < 0.001) and interaction effect ($F_{2, 21}$ = 5.3, $p$ = 0.014) (Fig. 4C, Table 2). *Post hoc* testing for the interaction effect showed that young visual normals had a 105%

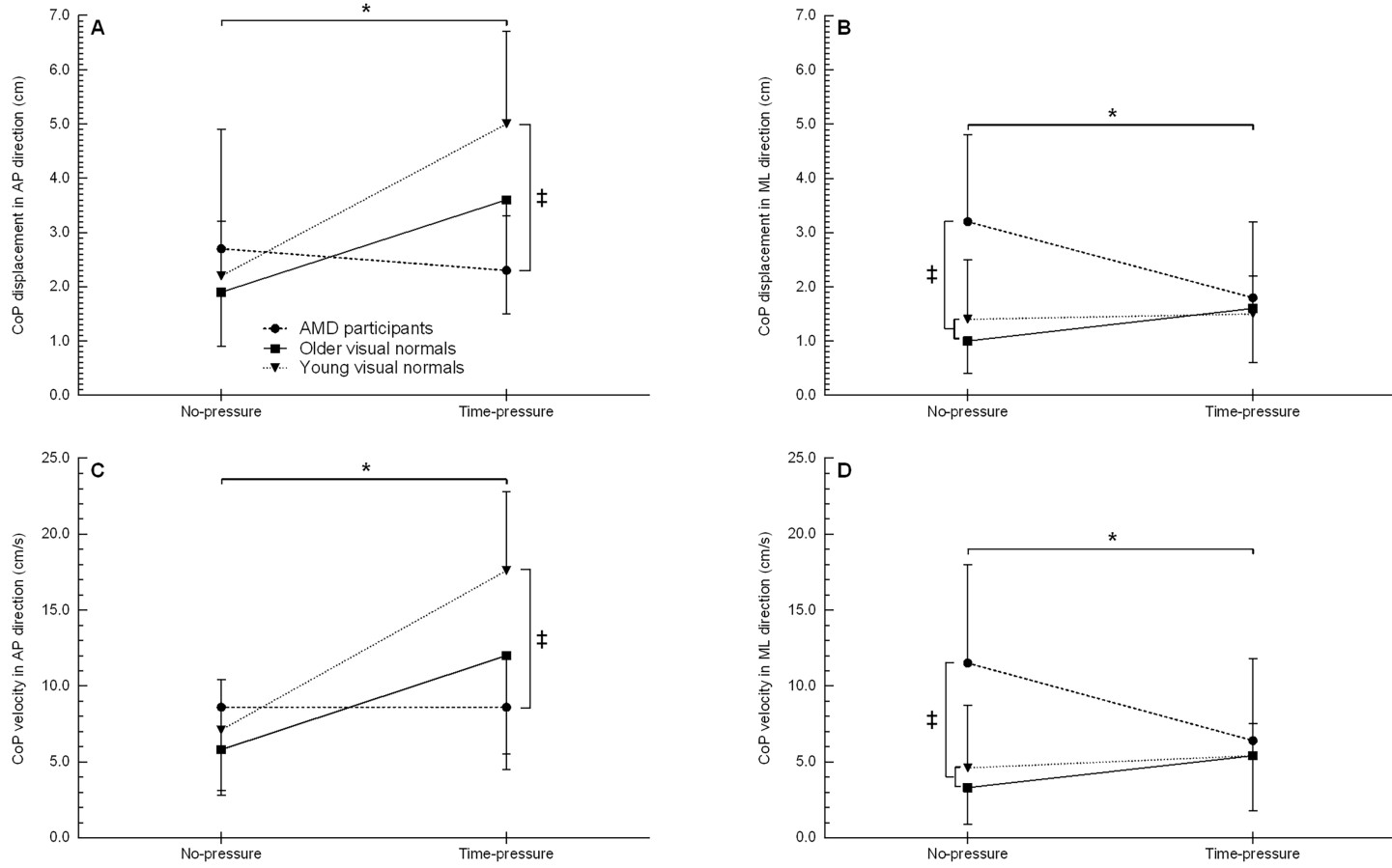

**Figure 4 Centre-of-pressure displacement and velocity when stepping up to the force platform with the trail foot in the time-pressure and no-pressure condition.** CoP displacement in the AP direction (A), CoP displacement in the ML direction (B), CoP velocity in the AP (C), and CoP velocity in the ML direction (D) in the two different conditions when stepping up to the force platform with the trail foot. AP, anterior-posterior; CoP, centre-of-pressure; ML, medial-lateral; ‡, between group difference ($p < 0.05$); *, group by condition interaction between the time-pressure and no-pressure condition ($p < 0.05$).

higher CoP velocity than AMD participants in the time-pressure condition ($p = 0.003$, d = 1.80). No other between group differences were observed (all $p \geq 0.051$). On comparing between conditions for each group, young and older visual normals showed a 108–149% higher CoP velocity in the time-pressure *vs*. no-pressure condition (both $p \leq 0.014$, d $\geq$ 1.18), while AMD participants did not significantly change their CoP velocity ($p = 0.995$). No significant main effect of group was observed ($F_{2, 21} = 2.8$, $p = 0.085$).

The average CoP velocity in the medial-lateral direction showed a group ($F_{2, 21} = 4.6$, $p = 0.022$) and interaction effect ($F_{2, 21} = 3.9$, $p = 0.037$) (Fig. 4D, Table 2). *Post hoc* testing for the interaction effect showed that AMD participants had a 60–71% higher CoP velocity than young and older visual normals in the no-pressure condition (both $p \leq 0.008$, d $\geq$ 1.20). No other between group differences were observed (all $p \geq 0.562$). On comparing between conditions for each group, AMD participants showed a 44% lower CoP velocity in the time-pressure *vs*. no-pressure condition ($p = 0.017$, d = 0.80), while young and older

visual normals did not significantly change their CoP velocity (both $p \geq 0.277$). No significant main effect of condition was observed ($F_{1, 21} = 0.4$, $p = 0.538$).

## DISCUSSION

Falls are common in older adults (*Rubenstein, 2006*), particularly when older adults have central field loss (*Szabo et al., 2010*; *Soubrane et al., 2007*). Stepping up to a new level is considered a challenging and hazardous everyday activity (*Startzell et al., 2000*) and older adults with AMD experience difficulties negotiating steps and stairs (*Taylor et al., 2016*; *Taylor et al., 2019*). Step negotiation in daily life (*e.g.*, stepping up a pavement curb) is often performed under time-pressure but the effects of time-pressure on step negotiation have not been investigated. The present study highlights how age and central field loss affect the movement kinematics and balance control when stepping up to a new level under time-pressure. The main findings reveal that despite walking faster, AMD participants retained a controlled landing strategy under time-pressure unlike young and older visual normals who demonstrated more forceful landing mechanics to maintain balance control during the step-up. Young visual normals showed the most forceful landing mechanics.

The loading rate in the vertical direction was affected by age, vision loss and time-pressure. Young and older visual normals increased their vertical loading rate under time-pressure while AMD participants did not (Fig. 3A). This result shows that AMD participants adopted a more cautious landing strategy under time-pressure compared to young and older visual normals. The more cautious landing strategy under time-pressure might be the result of poor oculomotor control and deficits in visual search behaviour that are observed in individuals with AMD (*Verghese, Vullings & Shanidze, 2021*). To illustrate, individuals with AMD demonstrated longer saccadic search latencies, more saccades, and longer fixation dwell times compared to individuals with normal vision (*Van der Stigchel et al., 2013*; *Boucart et al., 2015*), suggesting that individuals with AMD require more time to extract information from targets of interest. In addition, individuals with AMD experienced increased pursuit latencies compared to visual normals (*Shanidze et al., 2016*), implying that individuals with AMD need longer time to detect the target of interest and reach steady-state pursuit. In time-constrained situations, there is less time to detect targets of interest, reach steady-state pursuit, and extract information from those targets of interest. The more cautious landing strategy under time-pressure might help AMD participants to increase the kinaesthetic information in the absence of accurate visual information (*i.e.*, feeling the way down to the step rather than dropping onto it).

Compared to young visual normals, AMD participants and older visual normals also landed with less peak vertical force on the step. This finding corroborates that the amount of downward force applied during stair ascent decreases with age (*Stacoff et al., 2005*). It remains speculative why older adults apply less peak downward force but it is likely a combination of age-related functional decline (*Hebert, 1997*) and age-related deficits in the somatosensory and vestibular system (*Goble et al., 2009*; *Li et al., 2015*; *Hsieh, Lin & Lee, 2014*; *Baloh, Jacobson & Socotch, 1993*) that require a more controlled landing of the lead foot when stepping up a curb. The more controlled landing was not accompanied by a

trade-off in foot velocity, because the single support time of the lead foot during the step-up was not significantly different across groups.

Age and vision loss also affected the landing mechanics of the lead foot in the anterior-posterior direction, especially when the step-up movement was performed under time-pressure. As a consequence of completing the action faster under time-pressure (*i.e.*, shorter single support phase during the step-up of the lead foot), young and older visual normals recorded a higher anterior-posterior loading rate and ground reaction force while these landing mechanics did not increase in AMD participants. Interestingly, older visual normals and AMD participants spent 37–38% longer in the double support phase prior to the step-up than young visual normals, which allowed more time to acquire visual information for the control of adaptive locomotion (*Patla & Greig, 2006*). Visual information from the peripheral visual field is important for adjusting foot placement and lower limb trajectory when stepping up to a new level (*i.e.*, online feedback corrections), while the central visual field is important for feedforward planning (*Marigold, 2008*; *Patla, 1997*). The peripheral visual field tend to be generally intact in individuals with AMD and motion processing is preserved in the periphery. Nonetheless, the AMD participants changed to a more cautious landing strategy under time-pressure, suggesting that they did not receive sufficient sensory information to adopt a more forceful landing strategy like the young and older visual normals.

The somatosensory, visual, and vestibular systems are the sensory systems that play a role in step negotiation (*Startzell et al., 2000*). Ageing compromises the functioning of these sensory systems (*Goble et al., 2009*; *Li et al., 2015*; *Hsieh, Lin & Lee, 2014*; *Congdon et al., 2004*; *Baloh, Jacobson & Socotch, 1993*) and pathologies such as AMD can accelerate this deterioration (*Horak, Shupert & Mirka, 1989*). Individuals with AMD receive limited visual information from the central visual field and the functioning of the vestibular system is compromised compared to older visual normals, but the somatosensory system is intact (*Elliott et al., 1995*; *Radvay et al., 2007*). A compromised vestibular system also affects the coordination of eye and head movements in individuals with AMD due to deficiencies in vestibulo-ocular reflex cancellation (*Verghese, Vullings & Shanidze, 2021*). This cancellation is an important reflexive mechanism during locomotion to track and stabilize gaze on targets of interest (*Patla, 1997*). When locomotion is performed at a higher speed (*e.g.*, under time-pressure), there will be more head movement in the anterior-posterior direction, which places a higher demand on vestibulo-ocular reflex cancellation. Consequently, impairments in cancellation might become more apparent under time-pressure and AMD participants will acquire less accurate visual information from the central and peripheral visual field. The lack of visual information under time-pressure together with a compromised vestibular system, which is more challenged under time-pressure when the step-up movement is performed at higher speed, might explain why AMD participants adopted a more cautious landing strategy under time-pressure. The more cautious landing strategy allowed more input from the somatosensory system (*e.g.*, kinaesthetic information) and better balance control in the anterior-posterior direction.

The landing mechanics of the lead foot in the medial-lateral direction were affected by age and time-pressure but not by vision loss. The loading rate in the medial-lateral direction was lower in older visual normals and AMD participants compared to young visual normals, and only the young visual normals increased their medial-lateral loading rate under time-pressure. This finding supports the earlier idea that age-related functional decline and age-related deficits in the somatosensory and vestibular system require a more controlled landing of the lead foot when stepping up a curb. The medial-lateral loading rate at initial foot contact is important for balance control as it regulates the trajectory of the CoM so that it remains within the lateral borders of the base-of-support (*MacKinnon & Winter, 1993*). The higher medial-lateral loading rate for young visual normals suggests errors in foot placement due to the less controlled landing, especially under time-pressure. The errors in foot placement required a higher medial-lateral loading rate to prevent the CoM from traveling outside the lateral borders of the base-of-support (*i.e.*, initiating medial-lateral imbalance).

Balance control during the step-up of the trail foot (*i.e.*, single support phase) was affected by time-pressure and vision loss but not by age. Stepping up to a new level under time-pressure increased the CoP displacement and velocity in the anterior-poster direction for young and older visual normals but not for AMD participants. During the single support phase of the trail foot, the CoM is propelled forwards and the forward momentum needs to be counteracted to prevent a forward fall (*Winter, 1995*). All groups showed shorter single support times under time-pressure, which indicates that participants moved faster forward and that the CoM was traveling at a higher velocity. Greater muscular effort is necessary to counteract the forward movement of the CoM in order to prevent a fall forward (*Hahn & Chou, 2004*). The increased muscular effort causes a greater CoP displacement and velocity opposing the displacement of the CoM. Indeed, young and older visual normals increased their CoP displacement and velocity under time-pressure but AMD participants used a different strategy to counteract the forward displacement of the CoM. It might be that AMD participants adjusted their movement mechanics during step negotiation to keep the horizontal position of the CoM close to the centre of the base of support (*i.e.*, increased balance control), as seen in healthy older adults with simulated blur (*Heasley et al., 2004*).

Time-pressure and vision loss did also affect balance control in the medial-lateral direction during the step-up of the trail foot. In the no-pressure condition, AMD participants showed more CoP displacement and a higher CoP velocity (*i.e.*, reduced balance control) than the young and older visual normals. However, under time-pressure, AMD participant reduced their CoP displacement and velocity (*i.e.*, increased balance control) while young and older visual normals did not show a change in these balance control parameters. These results suggest that AMD participants altered their movement mechanics under time-pressure to also improve balance control in the medial-lateral direction. The reduced medial-lateral balance control of AMD participants in the no-pressure condition supports the findings of healthy older adults who stepped up to a new level with simulated blur (*Buckley et al., 2005b*). It might be that due to the nature of the task AMD participants adjusted their movement mechanics to not fall forwards,
thereby accepting more deviations in medial-lateral balance control ('minimal intervention principle') (*Todorov & Jordan, 2002*).

Age did not cause additional balance deficits when stepping up to a new level. The absence of age-related deficits in balance control during step ascent are in accordance with previous research (*Reid et al., 2011*; *Lee & Chou, 2007*). The present study is unique in measuring balance control in AMD participants during a dynamic task. Previous studies in AMD participants reported deficits in standing balance (*Wood et al., 2009*; *Elliott et al., 1995*; *Willis et al., 2013*; *Chatard et al., 2017*; *Kotecha et al., 2013*) but everyday activities are often dynamic rather than static. The dynamic situation in the present study did reveal deficits in medial-lateral balance control, but none of the AMD participants actually lost their balance during the step-up task.

It is important to highlight that fit and healthy individuals were recruited for the present study who did not have a history of falling. The present findings suggest a very positive picture for older adults with and without AMD; however, these population groups are at an increased risk of falling (*Szabo et al., 2010*; *Soubrane et al., 2007*; *Rubenstein, 2006*). Future research should consider recruiting vision impaired fallers to understand whether their balance and adaptive gait characteristics are different to the population group recruited to the current study. Also interesting is to look at vision metrics as predictors of balance and adaptive gait metrics, using a large dataset that includes a wide range of datapoints from no visual impairment to severe visual impairment.

### Limitations

In the current study, we were unable to measure the contribution of the trail foot towards maintaining balance control when stepping up to the floor-mounted force plate because only one force plate was used. The mechanics of the trail foot play an important role in maintaining medial-lateral balance control during a stepping movement (*Lyon & Day, 1997*), especially when vision was acutely impaired (*Buckley et al., 2005b*). Future research is needed to determine how the mechanics of the trail foot also help older adults with chronic vision loss to maintain medial-lateral balance control.

The step height in the present study was low (4.5 cm) compared to the step heights used in previous vision loss research (7.2 to 23.5 cm) (*Heasley et al., 2004*; *Buckley et al., 2005a*; *Buckley et al., 2005b*; *Timmis et al., 2014*; *Alexander et al., 2014*). These previous studies used step heights that people would usually encounter in buildings (*Billington et al., 2017*), while the step height in the current study was meant to reflect the height of a pavement curb. Pavement curbs at pedestrian crossing points can be as low as a few millimetres. In addition, curbs below the height of 6.0 cm are especially difficult for blind and partially sighted to detect the edge of the footway (*Thomas, 2011*). Following previous research (*Buckley et al., 2005b*; *Timmis et al., 2014*), it is expected that higher step heights will further emphasize the differences in landing mechanics and balance control observed in the present study.

## CONCLUSIONS

The present study demonstrates the effects of age and vision loss on landing mechanics and balance control when stepping up to a new level under time-pressure. Individuals with AMD did not adapt their landing mechanics under time-pressure (*i.e.*, remained more cautious), while the landing strategy of young and older visual normals became more forceful (*i.e.*, increased loading rates and ground reaction forces) with the young visual normals showing the most forceful landing. Retaining less forceful landing mechanics might be a safety strategy for older adults with and without central field loss to maintain balance control during adaptive gait, especially during time pressured situations when balance control in the anterior-posterior direction is more challenged.

## ACKNOWLEDGEMENTS

The authors thank the Macular Society and Cam Sight for raising awareness and assisting with the recruitment of participants for this research project and thank Cassandra Whelan for her assistance with the data collection and data processing. Lastly, the authors would like to thank Jonathan Allsop for his help with the laboratory setup to measure time-pressured adaptive gait.

### Funding

This work was supported by an Anglia Ruskin Fellowship. The funders had no role in study design, data collection and analysis, decision to publish, or preparation of the manuscript.

### Grant Disclosures

The following grant information was disclosed by the authors:
Anglia Ruskin Fellowship.

### Competing Interests

The authors declare that they have no competing interests.

### Author Contributions

- Tjerk Zult conceived and designed the experiments, performed the experiments, analyzed the data, prepared figures and/or tables, authored or reviewed drafts of the article, and approved the final draft.
- Matthew A. Timmis conceived and designed the experiments, analyzed the data, authored or reviewed drafts of the article, and approved the final draft.
- Shahina Pardhan conceived and designed the experiments, authored or reviewed drafts of the article, and approved the final draft.

## Human Ethics

The following information was supplied relating to ethical approvals (*i.e.*, approving body and any reference numbers):

The Research Ethics Committee of the Anglia Ruskin University granted Ethical approval to carry out the study within its facilities (approval number: FMSFREP 16/17 008).

## Data Availability

Raw data, including the baseline characteristics and task performance data of all participants, is available in the Supplemental Files.

## Supplemental Information

Supplemental information for this article can be found online at http://dx.doi.org/10.7717/peerj.14743#supplemental-information.

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
