# Peer review of "The effects of age and central field loss on maintaining balance control when stepping up to a new level under time-pressure"

_PeerJ, doi:10.7717/peerj.14743_

## Round 0.1 · original submission · Minor Revisions

Two reviewers have commented on your study, both agree that your study is of good quality and that building on previous findings it adds new information to the literature. The reviewers have some recommendations to further improve the quality of the paper.

·

Basic reporting

This is a well-written, clear manuscript with a thorough review of prior art in locomotor behaviors in simulated and central vision loss and clear explanation of the experiments performed and methods used.

One potential area for elaboration is the discussion of prior research with simulated blur when negotiating a step (~line 76 in Intro and further). I think there are two caveats that may be worth mentioning. First, the age of the cohort is important - how people adjust to both postural perturbation and sensory loss is age-dependent, and so discussing those results in the context of the age of the participant cohort may be warranted. The second limitation of that work is that blur of the whole visual field is not entirely equivalent to the loss of central field and development of eccentric fixation that occurs over time in macular degeneration. Additional reference is made to simulated vision loss on line 245. While the prior work is valuable and has obvious bearing on the current work, I encourage the authors to consider the limitations of how that work can be applied and the limitations of simulated vision loss.

A very minor comment is for line 92. "... when individuals have to attend their personal biological calls" is awkwardly phrase and also ambiguous. I think focusing on the time pressures of crossing the street or perhaps stepping up on an escalator step would be sufficient.

Experimental design

The experiment is well designed to answer the questions at hand. I had some comments that do not diminish the value of the study, but if answered could make the results more broadly and easily interpretable.

Is there a reason that the authors do not discuss task performance by the different groups? Did MDs complete as many of the timed trials as sighted older adults? Were they more likely to run out of time? What was the trial duration like for the different groups? Was there a significant difference?

In terms of matching participant groups, I would like to have known if there was any effort to match participants not only by age but by functional balance/fitness/habitual activity level? Those with MD are more likely to become more isolated and less active - this can be a potential confound in mobility studies. Was their fitness level assessed? For example, are MDs significantly slower when walking on the flat portion than MDs? Are they more prone to error? Was an assessment of functional balance or equivalent part of the screening? Further, the control group seems slightly younger, so it is not clear how well the age matching was done between the two older groups.

Finally, based on the results, it sounds like none of the groups had trouble with the obstacle. But is it possible that the obstacle affected time on task or other parameters to begin with? In other words, the three groups are all clearly able to cope with the obstacle and the main task, but is it possible that the metrics measured on task were also affected by the presence of the obstacle?

Have the authors considered looking at vision metrics (namely contrast sensitivity) as predictors of the balance metrics reported here? MD is highly heterogeneous, so perhaps a more per-participant approach might shed additional light on the interpretation of this data.

Minor question: which test was used to determine normality?

Validity of the findings

The findings are straightforward and interesting. I suggest that including more broad discussion of prior art regarding visual and oculomotor deficits in MD could strengthen some of the claims in the discussion.

I was hoping to see more of a discussion of the findings in the context of the authors' early reference to the increased tine demand for the visual inspection of the path by older adults and an even greater increase for those with MD. Prior work has shown increased oculomotor latencies and deficits in visual search in MD that may have direct bearing on locomotor tasks (particularly under time pressure).

On a related note, the discussion in lines 516-519 could use a more thorough review of what is known about visual search/information gathering in MD. I am not convinced that insufficient visual information is the only explanation of the more cautious landing strategy. What visual information is essential for optimal performance of this task? How might visual and oculomotor challenges in MD affect that? Presumably, they do not need to have examined the full scene and lower visual fields tend to be intact in MD and motion processing is preserved in the periphery. Conversely, what vestibular/postural/proprioceptive changes might also lead to the more cautious behavior observed?

Additional comments

Line 259 (Figure 1 caption): "lading" is probably a typo.

Discussion on lines 491-494 and Line 512 appears to be very similar both in verbiage and in the overall discussion of age and vision loss affecting landing mechanics. Perhaps the discussion could be consolidated across the different directions for a more complete summary of gait strategy,

Reviewer 2 ·

Basic reporting

Good.

Experimental design

Good.

Validity of the findings

Good.

Additional comments

This study investigated the effect of AMD visual impairment on stepping up to a new level under time pressures, compared to healthy young and older adult groups. The proposed study follows on from previous findings that adults with AMD are more cautious in their approach when stepping up and down a raised surface without time pressure. To advance previous findings, the authors aimed to determine whether time pressure (walking to the beat of an audible tone which increased in frequency) affected the stepping and landing action on to the raised surface. This small component is the novel aspect of the current study. The results indicate that older adults with AMD were more cautious, as hypothesised, and this is likely so that they maintain balance control. Overall the manuscript is well written, clear, and detailed. I do not have any major concerns but I have a few minor comments; most comments are related to clarity.

Introduction
The authors provide a good rationale for testing older adults with AMD under a time pressured situation. Further rationale could be provided to support why young and older adults with normal/corrected vision were included as comparison groups. One way to do this might be to add more information about the groups of participants previously tested in the studies reported in the paragraph starting on line 108.

The authors provide two hypotheses. One states the expected changes under no time-pressure (line 119) and one states the changes under time-pressure (line 125). The primary focus seems to be the effect of time-pressure on stepping in adults with AMD compared to healthy young and older adults, thus the hypothesis on line 125 seems most pertinent and could be the first hypothesis stated. The hypothesis relating to the no pressure condition stated on line 119 seems to be the reverse (something the reader can presume based on the time pressure hypothesis) and can be removed unless I am mistaken.

Methods
Please state the height, width and depth of the obstacle and raised surface.
Line 168 – the authors state that the raised surface was 4.5cm high, which is approximately half the height of the obstacle they had to negotiate prior to stepping up. At this stage in the manuscript I am not sure it is clear why the authors focus on the small raised surface (most curbs are ~10-15+ cm) and not the tall (10 cm) obstacle.
Line 190 – did the intermittent tone frequency increase as a function of the trial duration? Please clarify.
Line 193 – the participant walked 20% faster than comfortable speed by the end of the trial (not throughout the entire trial but at the point of approaching the step), is this correct? This could be clearer for future readers.
Why was a 20% increase in walking speed chosen?
Line 215 – how were gait events defined?
Line 229 – how was final foot placement defined temporally? “Touch-down” or “heel-strike” are typically used to determine the beginning of stance, was this the case here?
Line 237 – typo – “marker” rather than “maker”.
Line 239 – force thresholds were not used to determine gait events when landing on the force plate, why were kinematic events preferred?
Line 243-246 – the authors clarify why the obstacle data was not analysed in this study but it would be nice to know whether the older adult AMD data confirmed what was previously reported when simulating vision in Zult et al 2019.
Line 259 – type – “landing” rather than “lading”.

Discussion
Line 501-504. It would be nice to know how foot velocity changed during the process of stepping on to the raised surface. It may be that there is a trade-off in foot velocity to ensure foot placement accuracy and precision when stepping/landing on to the raised surface.
Line 581 – the limitations section is clear and informative, but I wonder if the authors considered adding more justification for choosing a step height of 4.5cm to the methods section of the paper. This could curtail the concern of future readers who might not feel an increased surface height of 4.5cm is sufficient.
The authors may like to consider using the margin of stability as a measure of balance control for future similar research as a means of assessing the centre of mass position and velocity relative to the centre of pressure, though this would require a more in-depth marker set to be applied.

---

## Round 0.2 · accepted · Accept

The authors have carefully addressed the reviewers' comments and the manuscript is now ready for publication.

·

Basic reporting

Good

Experimental design

Good

Validity of the findings

Good

Additional comments

None

Reviewer 2 ·

Basic reporting

Very good throughout.

Experimental design

Very good throughout.

Validity of the findings

Very good throughout.

Additional comments

The authors have addressed all my comments from the first review and provided sufficient detail. Well done on producing this piece of work.